# Epitaxial Growth of Silicon on Silicon Wafers by Direct Laser Melting

**DOI:** 10.3390/ma13214728

**Published:** 2020-10-23

**Authors:** Marie Le Dantec, Mustafa Abdulstaar, Marc Leparoux, Patrik Hoffmann

**Affiliations:** Empa–Swiss Federal Laboratories for Materials Science and Technology, Laboratory for Advanced Materials Processing, Feuerwerkerstrasse 39, CH-3602 Thun, Switzerland; marie.ledantec@empa.ch (M.L.D.); mustafa.abdulstaar@gmail.com (M.A.); Patrik.Hoffmann@empa.ch (P.H.)

**Keywords:** direct laser melting, additive manufacturing, silicon, epitaxial growth

## Abstract

Additive manufacturing (AM) of brittle materials remains challenging, as they are prone to cracking due to the steep thermal gradients present during melting and cooling after laser exposition. Silicon is an ideal brittle material for study since most of the physical properties of single-element materials can be found in the literature and high-purity silicon powders are readily available. Direct laser melting (DLM) of silicon powder was performed to establish the conditions under which cracks occur and to understand how the solidification front impacts the final microstructure. Through careful control of process conditions, paying special attention to thermal gradients and the growth velocity, epitaxial pillars free of cracks could be grown to a length of several millimeters.

## 1. Introduction

Powder bed laser scanning, particularly selective laser melting (SLM) and selective laser sintering (SLS), are among the most widespread additive manufacturing techniques. Both SLM and SLS are routinely used in industry to manufacture metal parts with complex geometries for rapid prototyping and to produce custom objects where low part counts preclude the use of dies or other mass-production processes. Two main drawbacks of SLM and SLS are the time-consuming powder bed preparation and the need to recover and recycle unused powder. Both of these disadvantages can be overcome by forgoing the powder bed and, instead, simply adding material in powder or wire form directly to the melt pool. This approach is closely related to laser cladding and includes laser metal deposition (LMD), laser engineered net shaping (LENS) and direct energy deposition (DED), among others. Using these approaches, it is even possible to successfully repair or modify existing parts, overcoming the big challenge of avoiding solidification shrinkage and hot cracking for high-performance alloys.

Additive manufacturing of polymer materials usually results in an amorphous or semicrystalline material deposit, whereas inorganic dielectrics are mainly deposited as mixed oxides, ceramics or glasses. The laser-induced deposition of brittle, monocrystalline, dielectric material was carried out for sapphire [1] based on an older work from 1972 [2]. Wilms et al. also reported successful single-crystal alumina production using LMD [3].

An extensive body of literature concerns epitaxial growth of metals through additive manufacturing, which was well summarized in 2016 by Basak and Das [4]. Scientific contributions concerning the epitaxial growth of pure metal elements from powder precursors by laser are very rare. However, electron beam melting (EBM) of pure metal powders resulted in epitaxial growth of Cu, Nb and Fe. In the same review article, epitaxial growth of prealloyed material systems was presented, with a focus on Fe-based, Ni-based and Ti-based alloys. A significant amount of work has also been done on Al-based, Co-based and Cu-based alloys.

Epitaxial growth of silicon on Si wafers using a powder-based additive manufacturing process has not been reported so far in the literature. For this short innovation paper, we focused on powder-feed laser printing (also known as direct laser melting or DED) of silicon, a brittle elemental material with properties contrasting strongly with those of metals and polymers, which are the subject of most of the available literature. In considering the melting and solidification of silicon, some properties are especially noteworthy. Silicon’s liquid density is 10% higher at the melting point than its density in its solid form, i.e., its behavior is anomalous. Like ice on water, solid silicon will float on liquid silicon. Furthermore, silicon is the element with the second highest molar melting enthalpy, surpassed only by carbon.

Silicon is the ubiquitous material used in nearly all integrated circuit (IC) devices, micro-electromechanical systems (MEMS) and further developed devices such as MOEMS and NOEMS (Micro- and Nano-Opto-Electro-Mechanical Systems), produced in cleanrooms. A broad suite of deposition, modification and removal processes for silicon have been developed, including lithography, chemical etching, plasma etching, chemical vapor deposition (CVD), physical vapor deposition (PVD) and nearly all standard processes adopted from classical metallurgy.

As a single-element pure material, silicon is ideal for a more fundamental study of crack formation and grain growth in 3D printing, especially for materials that also grow with a faceted liquid/solid interface. Free-form 3D silicon printing has been attempted using a number of different focused beams, including lasers [5,6] and electrons [7]. The raw material can be deposited directly or from chemical or gaseous precursors like silanes. In the 21st century, additive manufacturing processes, such as powder bed laser melting, have been increasingly studied. However, only one paper was found on powder bed 3D silicon printing. This work focused on the production of silicon nitride end devices [8]. Silicon powder was mixed with nitridation catalysts and “printed” by SLM. In this study the emphasis was on part density rather than microstructure, leaving room for further study.

A wide range of potential uses for additively manufactured silicon parts is available, ranging from structural to electronic applications. High aspect ratio epitaxial structures and free shapes are ideal; they resist high-cycle loading and low defect densities prevent fatigue. Silicon has already supplanted metal as the material of choice for the mainspring in mechanical watches, although these structures are currently produced with cleanroom techniques. Additive manufacturing offers new design possibilities for complex part geometries and avoids material wastage and complex, time-consuming, cleanroom processing. In the short term, additive manufacturing (AM) processes are unlikely to compete directly with lithographic parts in terms of quality and dimensional tolerances, but reduction in powder size, the use of microfocus lasers and more precise stage motor control offer hope for quality improvement. The present study demonstrates, for the first time, the epitaxial growth of millimeter-diameter silicon pillars on silicon wafers from silicon powder with 10-µm diameter particles and a near-infrared pulsed Nd:YAG laser as a heat source.

## 2. Materials and Methods

The experimental setup employed in this study used direct laser melting (DLM) and was presented in detail in previous publications [9,10]. It is shown schematically in Figure 1. Only the important features are presented briefly in this section. We used a LASAG SLX 200 flash lamp (Coherent Switzerland AG, Belp, Switzerland) and a pumped Nd:YAG laser emitting adjustable microsecond-long pulses at a 1064-nm wavelength as a heat source. In this study, the laser pulse was constant at 1 ms. The laser light was transported from the laser to the laser head with a 400-µm inner core diameter step index fiber LL422 (Coherent Switzerland AG, Belp, Switzerland). The laser light was focused with an optical head to a spot size of about 600 µm on the silicon wafer substrate through a glass window with a large depth of field of about 4 mm. The spatial beam profile was a conically topped flat hat. Pulse energies from 160 to 405 mJ were applied (measured intensities that hit the wafer surface), corresponding to average laser intensities during the pulses of 160 to 405 W, respectively. Laser intensities of 26.7 to 67.5 MW/cm^2^ were obtained. Preheating of the wafer was possible using a ceramic covered resistive heating plate (Heizelement-LB 7E-10000-17-13, Bach Resistor Ceramics GmbH, Werneuchen, Germany) prior to the laser pulses to overcome the brittle-to-ductile temperature of silicon [11,12,13]. The indicated temperatures are the ones measured at the top surface of the wafer with a thermocouple glued with a ceramic paste. The temperature of the substrate was maintained at 730 °C (±3%) during the whole deposition process to exceed the brittle-to-ductile transition of silicon without damaging the heating plate and the stages. 

The growth process was realized by moving the substrate downwards, away from the laser focal point, in order to keep the laser spot aligned with the growing tip of the deposits at speeds between 0.0167 to 0.3 mm/s. For the silicon supply, commercially available pure silicon powder (Dahlian King Choice, Dalian, China) was used. The powder had a large particle size distribution and flowed poorly because of the sharp edges and irregular shape of the particles. The particle size of the powders was measured by image analysis (Powdershape, IST AG, Vilters, Switzerland) and was the following: d_10_ = 13.0 µm, d_50_ = 43.1 µm and d_90_ = 93.4 µm. Its purity was measured by glow discharge mass spectrometry (GDMS) and was 4N. To overcome the poor flowability, the powder was transported with a commercial pulsed powder transport system (Impakt, P&S Powder and Surface GmbH, Salzkotten, Germany), which is based on cell evacuation, powder sucking, compression and release cycles. The resulting powder pulses of approximately 0.1-s duration had a frequency of 5 Hz. This resulted in a duty cycle of gas and powder supply of 50%.

Pillars were grown perpendicular to the wafer substrate surface by pulsed spraying of the silicon powder from the side at an angle of about 50° from the wafer surface. The powder supply (powder feed rates from 1 to 47.5 g/min) was adjusted with the z-axis moving speed in order to obtain more or less cylindrical pillar deposits grown perpendicular to the wafer substrate. The powder jet was directed to the focus of the laser using a simple pipe with a 3-mm internal diameter that was much larger than the laser beam diameter (600 µm).

After production, the pillars were embedded in a resin (7x SpeciFix Resin and 1x SpeciFix curing agent, Struers, Kemen, Germany) before grinding and polishing. Manual grinding was necessary to avoid the formation of polishing cracks. A grinding wheel and silicon carbide abrasive paper (WS FLEX 18 C, Hermes Schleifmittel GmbH, Hamburg, Germany) were used (grids ranging from 180 to 4000). Slurries made with 1-μm diamond particles followed by a polishing emulsion containing 0.04-μm silicon dioxide particles were used to manually polish the surface. The samples were subsequently cleaned with water and ethanol and dried using a pressurized air stream. They were first analyzed by optical microscopy in order to observe if there was any cracking. Characterization of the crystalline orientation of the grown material with respect to the wafer was then implemented by an electron backscatter diffraction (EBSD) detector (TSL, TexSEM Laboratories, Inc., Provo, UT, USA) mounted in a scanning electron microscope (SEM) chamber (DSM 962, Zeiss, Munich, Germany). The EBSD maps were obtained with an acceleration voltage of 15–20 kV and a beam current of about 3 μA, at a working distance of 14–16 mm, with a step size of 2–7 µm. The EBSD images are always presented with their crystallographic orientation in the direction perpendicular to the wafer surface. More details are available in Le Dantec’s PhD thesis report [9].

## 3. Results

### 3.1. Attachment to the Substrate

Silicon pillar growth occurred in the direction of the impinging laser and powder supply over a wide range of laser and powder feed conditions if the substrate displacement speed was roughly adapted to the powder feed rate. The initial stage of growth was very critical as the appearance of cracks resulted in the substrate material close to the additively manufactured material detaching the latter from the substrate. Figure 2 highlights two combinations of laser frequency and energy per pulse (50 Hz/405 mJ and 200 Hz/160 mJ) used with both unheated substrates and preheating to 730 °C. Detachment of the pillar from the substrate occurred when the substrate was at room temperature (Figure 2a,c), crack formation occurred in the case of a preheated substrate at 50 Hz/405 mJ (Figure 2b) and crack-free attachment was obtained for 200 Hz/160 mJ when the substrate was preheated to 730 °C (Figure 2d). The detachment or cracks appeared due to thermal stresses induced in the silicon substrate. Successful attachment of the growing silicon pillar to the substrate was obtained if two conditions were satisfied: when the substrate was preheated to 730 °C in order to overcome the brittle-to-ductile transition of silicon and when the laser pulse repetition rate was sufficiently high (above 100 Hz for a feed rate of 15 g/min). Under these conditions, excessive thermal gradients around the melt pool were avoided. Furthermore, we observed that decreasing the powder feed rate also decreased the threshold frequency of the laser repetition rate for crack formation down to 75 Hz at 1 g/min. The asymmetric and poorly controlled shape of the pillar was mainly due to the radial injection of powders and the large and divergent powder jet, which induced a low catching efficiency.

These results were discussed in more detail for laser spot melting of silicon and the addition of powder to the melt pool in Le Dantec’s PhD thesis [9].

### 3.2. Epitaxial Growth

After establishing the process window under which no cracks occur, it was possible to focus on controlling the microstructure of the pillars themselves. Pillars were grown under various feed rates ranging from 1 up to 47.5 g/min. The adjustment of the vertical substrate speed and the powder feed rate was an important consideration in order to maintain a constant temperature gradient across the liquid/solid interface throughout pillar growth. At the base of the pillar, the substrate itself acted as a large heatsink from which the melt pool became increasingly thermally isolated as the length of the pillar increased. At low powder feed rates (and correspondingly low stage speeds), the growth of the pillars was observed to be consistently epitaxial. The silicon solidified with the same crystal orientation as the original silicon wafer substrate. We performed epitaxial growth beyond 1 mm in height from the wafer, corresponding to a height-to-diameter aspect ratio of 2 or more without extensive optimization of the parameters. Some twin grain boundaries were observed, but these were considered as part of the epitaxial front since they were not detrimental to the electronic properties of the material and, in fact, might provide enhanced hardness and toughness. The experiment was repeated several times within the same range of conditions (lowest powder feed rate of 1 g/min and various low stage speeds between 0.0083 and 0.1 mm/s) to demonstrate the reproducibility of these results. Additional EBSD maps of pillar growth experiments under different conditions can be found in Le Dantec’s PhD thesis [9]. Most of the tested growth experiments were carried out with Si <100> wafers. Fewer experiments with <111> wafers revealed that growing the <111> orientation resulted in less twinning, but in our tests, the epitaxial growth ceased closer to the substrate. Figure 3 shows the growth of pillars grown on two different substrate orientations. Figure 3a shows the growth on a <100> wafer and Figure 3b shows the growth on a <111> wafer. In Figure 3a, on the right side of the pillar, it can be seen that silicon drops off the melt pool along the pillar side during the process. In Figure 3b, twins appear as red and light blue zones and are considered to belong to the same epitaxial front. Increasing the powder feed rate to 15 g/min and keeping the other conditions constant resulted in increased diameter of the grown silicon and reduced the height of epitaxial growth (see Figure 3c).

The raw powder purity also played a role, as lower purity resulted in more grain nucleation and inferior epitaxial growth [9].

## 4. Discussion

The pillar geometry was determined by the size of the melt pool and by the pulsed supply of silicon powder impinging obliquely onto the substrate. Under pulsed laser irradiation conditions during growth of one pillar, the melt pool was small at the beginning of the growth, due to the efficient cooling of the substrate acting as a heat sink. As the solidification front moved further away from the substrate, heat flow was restricted through the base of the pillar, resulting in a larger melt pool and an increase in pillar diameter from 100 to 500 µm. For this first study, no modification of the pulsed power supply or energy input per time was undertaken. The energy per pulse and the pulse rate of the laser were also kept constant for all experiments; therefore, all pillars produced showed an increased diameter as their height increased (see Figure 2 and Figure 3). The liquid/solid interface line during growth moved upwards, away from the substrate. In the liquid region, just above the solidification line, solid powder particles from the powder supply attached without being completely molten, acting as nucleation sites for crystals growing radially in the cases of low powder supply (Figure 3a,b). This decreased the epitaxial growth of the full pillar. With increasing height, this effect was reinforced due to an increasing contribution of radial heat flow with respect to the axial heat flow. At a higher powder feed rate, as shown in Figure 3c, the amount of silicon powder added per time reaches a level where only partial melting of all the grains takes place, which might result in the observed polycrystalline silicon. Other effects influencing the stability of the epitaxial growth process included spattering or ploughing due to the impact of larger solid powder particles into the melt. With an insufficient stage speed or an excessive powder feed rate, large melt pools may form atop the pillars, spill over the perimeter and run down the pillar. The pillar in Figure 3a exhibits this behavior. Impingent particles can become trapped in the molten spillover, introducing an undesired secondary nucleation site. Process optimization is currently underway to prevent these types of defects from occurring and to achieve smaller feature sizes.

When the powder impinges on the melt pool, the particles are wetted. Due to the anomalous 10% greater density of liquid silicon as compared to solid silicon, the solid particles float on the liquid silicon surface until they melt. During the initial phase of growth, the liquid silicon at the bottom of the melt pool was in contact with the wafer that acted as a heat sink. Therefore, the thermal gradients (G) at the liquid/solid interface were high and perpendicular to the wafer surface. With a relatively low feed rate and stage speed resulting in a relatively low growth rate (V), the ratio G/V was high enough to ensure epitaxial growth [14,15]. Moreover, a small feed rate decreased the risk of insufficient particle melting which created potential nucleation sites [16]. Due to the (10%) increase in volume during solidification of pure silicon, no hot solidification cracks, such as those known to occur in brittle alloys due to solidification shrinkage, seemed to appear in these experiments. Therefore, additive manufacturing with this anomalous material should be the subject of further in-depth investigation in the form of systematic studies.

The process is not yet sufficiently mature to create more complex geometries. Defect analysis and dislocation density calculations need to be carried out. Improved pillar quality with tighter geometric tolerances while maintaining epitaxial growth should be achievable through improved heat flux control and by modifying the powder supply system and stage speed.

## 5. Conclusions and Outlook

We reported the first successful epitaxy of Si from silicon powder onto silicon wafers in a DED process. The crystalline orientation of the wafer was maintained throughout the pillar height as demonstrated by EBSD mapping of the prepared cross-sections of the 1-mm diameter and several millimeters high pillars, without laborious fine-tuning of parameters. Crack-free deposition turned out to be more challenging than epitaxial growth, the former due to the high brittleness of cold silicon. Modification of the powder supply geometry, reduction of the laser spot size and reduction of powder particle diameters, together with systematic studies of the parameters, will pave the way for our industrial partners’ applications in MEMS technology and the watchmaking industry. Within this scope, advanced manufacturing of small functional structures of metals or silicon with wall width or wire diameter dimensions below 100 µm is currently not easily achieved. Light focusing to less than 20 µm with precision stages of 3 µm in space is already realized, as are accurate axis movement and high-resolution real-time observation systems. Nevertheless, in this contribution, several aspects of the direct metal deposition 3D printing process were addressed and the epitaxial growth of Si on silicon wafer substrates was demonstrated as a first step. Miniaturization and control and the mastering of deposit geometry, deposit surface roughness, dimensional precision and process efficacy represent some of the many more areas that remain to be addressed. One of the major challenges we are currently facing is sustainable (low-waste), precise, high-resolution material transportation into tiny laser melt pools.

## Figures and Tables

**Figure 1 materials-13-04728-f001:**
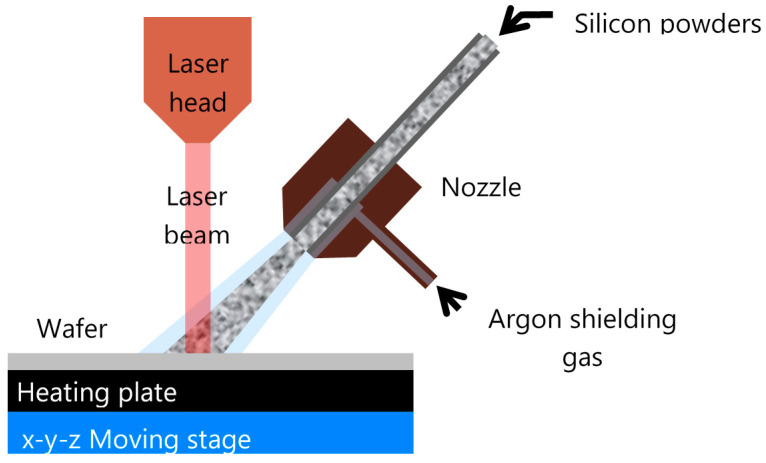
Schematic description of the process [9].

**Figure 2 materials-13-04728-f002:**
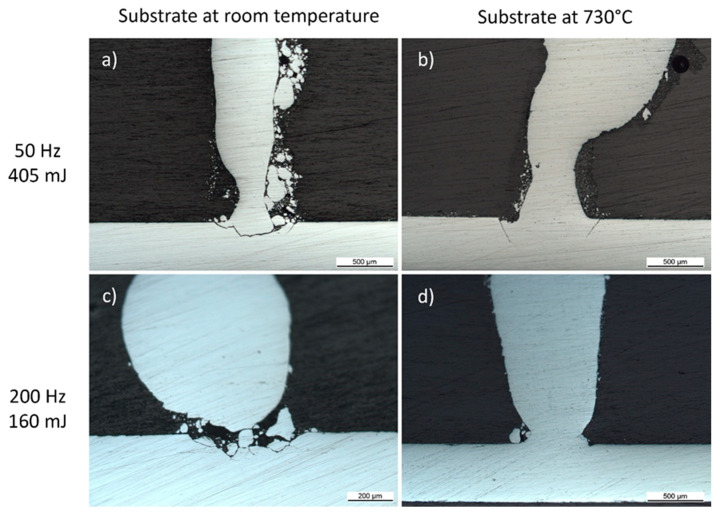
Cross-sections of pillars built on Si <100> wafers using a 1-ms pulse length and a powder feed rate of 15 g/min as a function of different input energies and substrate preheating above the brittle-to-ductile transition of silicon [9]: (**a**) room temperature (RT), 50 Hz, 405 mJ; (**b**) 730 °C, 50 Hz, 405 mJ; (**c**) RT, 200 Hz, 160 mJ; (**d**) 730 °C, 200 Hz, 160 mJ.

**Figure 3 materials-13-04728-f003:**
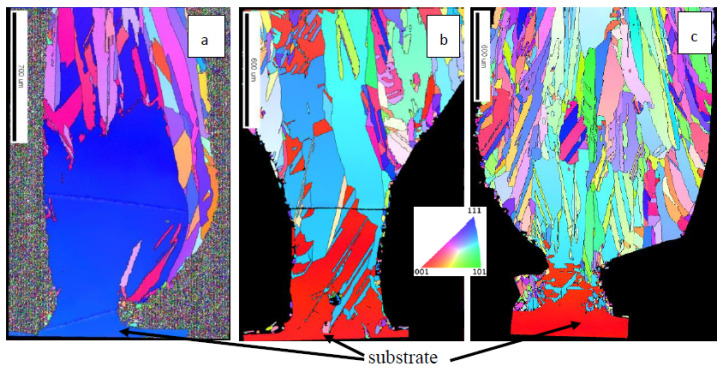
Electron backscatter diffraction (EBSD) mapping of pillars built with pulse energy of 161 mJ, laser pulse duration of 1 ms, stage speed 0.0167 mm/s, frequency of 200 Hz and powder feed rate of 1.0 g/min on a Si <111> wafer shown in dark blue color (**a**) and on a <100> substrate shown in red color (**b**) A higher powder feed rate (15 g/min) with otherwise identical conditions resulted in polycrystalline growth (**c**). The inverse pole figure (IPF) color scale shows the crystal orientations in the pillar growth direction [9]. The corresponding SEM pictures (**d**–**f**), including the marked grain boundaries, reveal some deposition defects, post-deposition sample preparation cracks and polishing scratches.

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
