# Peer review of "Epitaxial Growth of Silicon on Silicon Wafers by Direct Laser Melting"

_materials, 2020, doi:10.3390/ma13214728_

Round 1
Reviewer 1 Report
In the present study, the authors reported the successful epitaxy of silicon from Silicon powder onto Silicon wafers in a direct energy deposition process. The crystalline orientation of the wafer is maintained throughout the pillar height as demonstrated by electron backscatter diffraction mapping of the prepared cross-sections of the 1 mm diameter and several mm high pillars. The manuscript is well organized and contains some interesting findings. Thus, I recommended a major revision of the article from its present form before it can be published in materials. The main concerns are listed below.
- The abstract and conclusion sections should be a specific and scientific approach.
- The authors should explain the novelty, need, and importance of the present work?
- The motivation part is a lack of introduction. The authors should revise the introduction part.
- The schematic description of the experimental process is unclear.
- The substrate is preheated to 730°C. The authors should explain in detail why they choose this particular condition.
- The authors should provide the XRD and Raman spectra.
- The analysis of the defects and dislocations density has been playing a crucial role in the growth mechanism. The authors should provide data.
- Grammar and spell errors existed in the manuscript. Therefore, the authors are advised to recheck the whole manuscript for improving the language and structure carefully.
Author Response
We thank the reviewer for his improvement suggestions. Below we tried to answer point-by-point your comments
- The abstract has been reformulated. We also agree that the conclusion was probably too short and we thus developed it further combined with the outlook part.
-
We decide to publish a communication paper because this is the first reported epitaxial growth of silicon from powder in a laser based additive manufacturing. Of course a lot of optimization and additonal experiments are necessary to fully assess the potential of this result but first discussions with watch and MEMS manufacturers encourage us to continue this activity.
- Initially the motivation was to achieve something that nobody had done before without having fully identify industrial applications. Of course with our background in microelectronics and clean room processing, we found interesting to build up monocrystalline silicon structures with additive manufacturing instead of using a subtractive techniques inducing a lot of wastes. A sentence has been added in the introduction.
-
As the results presented here are unique and reported for the first time in the additive manufacturing community, we wanted to publish them as a short communication paper. In this format we are restricted and cannot provide as many details as in a normal paper. We refer to a public available PhD work where the methods and devices are well described.
-
This temperature value has been chosen to be above the brittle-to-ductile transition of silicon but the heating plate limited us also and the stages could have been damaged at temperatures above 80°C according to the specifications. We add a sentence in the manuscript.
-
Indeed Raman and XRD will be interesting to understand and characterize the stress. For this preliminary study, we just characterized the microstructure using EBSD to prove the epitaxial growth and as the cracks formation could be managed by heating up the substrate, we did not investigate it further. This will be performed and published in a further paper together with additional mechanical performances.
-
As mentioned above, this is a first communication of successful epitaxial growth of Silicon on silicon wafers. We agree with the reviewer that defects, dislocations and many more influencing parameters play crucial roles but have not been investigated in detail yet. We modified the text in the introduction and in the conclusions in order to explain the first successful epitaxial growth of Silicon message of this contribution.
- We hope the improvement of the manuscript makes it clearer. English has been checked again.
Reviewer 2 Report
please find the attached report.

Author Response
- We thank the reviewer of sharing our thought that printing single crystal silicon on silicon wafer is novel and interesting. For both reasons we prepared this short communication paper. We tried to express this fact more clearly in the introduction section.
- From our industrial partners from the MEMS and watch making industry an interest in 3-D printing of crystalline silicon on silicon was mentioned being of interest, but detailed applications of 3-D printing of epitaxial Silicon on Silicon were not divulged to us. Knowing very well that the dimensions and precision needs further improvement, we state again the aim of this contribution is to present first successful growth of Si on Si showing that it is possible, without having completed all the way of optimization, miniaturization and process mastering.
- In this study we used a large laser spot of 600um diameter that allows easy high speed camera observation of the process, therefore changing conditions during processing and observing growth is simpler than doing design of experiments without knowing any parameter values to start with. The results were obtained without any large systematic parameter optimization, this will be carried out when smaller spot size, higher precision and better process control will be aimed at. Growth obtained over quite wide range of parameters, epitaxy appears quite easily. Final applications such as watch parts or MEMS will need much smaller dimensions, which requires smaller spot size, smaller particle diameter powder, and process parameter search based on the present study that is ongoing at present.
- Supporting references for 3-D printing of Silicon or other brittle semiconductor materials are already all cited – surprisingly this is still a virgin field, therefore not many references can be given.
- Novelty is that the presented results are the first ever reported epitaxial growth of silicon on silicon. The main result is that it is possible to deposit silicon on silicon with pulsed laser and pulsed powder supply if the substrate is preheated above the brittle to ductile transition temperature of Silicon. There are no other bullet points to be addressed.
- We did not plan to give a trend but we, as far as we know, are the first one reporting this epitaxial growth with a laser assisted additive manufacturing process. We also select to write a communication paper for a short description of the results. Much more details about the process can be found in the reference 9.
Reviewer 3 Report
This manuscript mentioned that an attempt of epitaxial growth of Si wafers by selective laser melting of Si powders. This trial is interesting. However, there is numerous lack of unclear explanations. The manuscript needs to be major changed for publication.
- English may be improved. For example, in L.78, “measured intensities that “hit” the wafer surface”. “exposed” or “irradiated” is appropriate.
- What’s the absorbance of the exposed laser pulses to the Si powders? I think the reflectance of the Si powder is not so low. In addition, the absorption must depend on the feeding system geometrically because the Si fed to the wafer could disturb the exposure of the laser pulses.
- The repetition rate of the laser pulses seems to be important to avoid cracks induced by strong cooling of the silicon. I wonder that the pulse duration is also important to discuss the thermal history of Si because the duration between pulses is the same order of the pulse duration (~millimeter seconds, mentioned in L.72). What’s the pulse duration of the laser used in this research?
- The controlling the shape seems to be difficult in this process as shown in Fig.2. Is there any idea to improve the controlling? Why did the authors decide the spraying angle of 50°?
- By comparing the diameter of the powder to the laser beam spot, the beam spot was approximately 5 to 50 times larger than the diameter. If you explain the strategy to decide the sizes of the diameters of the powders and the laser beam spot, it must be helpful for the readers.
- In the EBSD mappings of a pillar, the core of the pillar was epitaxially grown. However, the diameter of the epitaxial front decreased far from the Si wafer. What was dominant for the epitaxial growth, heat temperature or heat duration? If the fabricated pillar was heated in an oven, the phase in the epitaxial growth increased? I wonder the mechanism is important to improve the epitaxial growth in laser processing because the pulse duration and repetition rates must be determined by considering those points.
Author Response
We thank the reviewer for his valuable scientifc comments. Beside the modification of the manuscript, we give point-by-point explanation in the following:
English has been checked again
The pulsed powder feed system has a duty cycle of about 50% and the powder was flowing out at low volume fraction in a strongly diverging powder stream. The particle laser interaction is therefore relatively small and effecting the laser exposure, therefore the optical interaction of the powder with the laser light can be neglected in this experimental set-up. We agree that the light reflection and probably even more light scattering will influence the process with higher duty cycles of the powder supply or even more with more concentrated powder streams, but not in our present conditions. The powder feeding angle will also influence strongly the light substrate interaction, but as mentioned above, with so diluted powder flows it was not relevant.
The pulse duration of 1 millisecond was fixed in this presented study as stated in the paper, yes, the pulse duration and the pulse energy together do evidently determine the heat input and therefore guaranty together the melt pool. These facts are described in more detail in the PhD thesis [ref 9] and in a paper about melting of silicon by pulsed laser melting that is under preparation but not submitted yet.
For a first feasibility of growing Silicon epitaxially, we did not try to improve any geometrical aspects, better control of growth geometry as well as many other parameter optimization is presently under investigation.
The powder diameter was determined from commercially available pure Silicon powder, and the laser spot size was selected much larger than the particle diameter to avoid too strong process impact by the pulsed powder supply (see answer above), and large in order to observe easily during parameter search. For process optimization both have to be optimized and depend one on the other.
We agree with the observation in the EBSD maps and simulated the heat flow during growth. Indeed radial heat flow increases with respect to the dominant axial heat flow with increasing height of the pillars and might help to explain the decreasing epitaxially grown diameter with increasing height, but other more technical less scientific parameters such as decreasing powder flow density at increased distance from the substrate, badly synchronized z-axis displacement speed differing from growth rate, and other factors such as small unmolten particles attaching to the molten walls might also influence this change. We did not carry out any post treatment processes of the pillars, because we believe that optimization of the parameters should result in improved growth quality without post treatments.
Reviewer 4 Report
The manuscript entitled “Epitaxial growth of silicon on silicon wafers by Direct Laser Melting” is beneficial for researchers working on this alloy and could also be useful for researchers in the Additive Manufacturing area. However, I suggest putting more references for the facts they mentioned in introduction section.
Author Response
We thank the reviewer for the proposition of improvement and have the impression that this reviewer understood the "letter" type "novelty announcement" of the manuscript, whereas pure Silicon we would not call it an alloy.
Due to the uniqueness of this work, it is difficult to find published relevant papers for comparison. We then decided not to put additional references in this short communication papers.
Round 2
Reviewer 1 Report
The manuscript is well organized in the revised version and it can be accepted in the present form.
Author Response
we thank the reviewer for accepting our paper.
Reviewer 2 Report
Please find the attached report.

Author Response
Comments on the revised version of “materials-886898”
Title: “Epitaxial growth of silicon on silicon wafers by Direct Laser Melting”
- There are a lot of question marks, and comments to be clearly answered in the revised manuscript, this will help to avoid any distraction by the reader and justify the presented resulted to be ready for publication:
1- First of all; apart from the type of publication presented (communication/ research article or letter), the scientific results should be validated clearly using a minimum of two independent experimental methods in addition to strengthening the novel contribution from the literature findings.
We do not agree with the reviewer that scientific results have to be validated using a minimum of two independent experimental methods. If the selected method is a clear, very well calibrated method such as Rutherford Backscattering Spectrometry (RBS) or Electron Back Scatter Diffraction (EBSD) with unambiguous results, the measurement with X-ray diffraction or other means of individual grown pillars do not make sense.
2- The authors stated that; “We reported the first successful epitaxy of Si from Silicon powder onto Silicon wafers in a DED process.” this point was not achieved inside the results and discussion sections. The characterization of a successful layer attachment to the substrate could not be validated by only a microscopic analysis, what about the adhesion area characteristics between the deposited layers and the substrate plate? What are the mechanical properties such as microhardness values? The scale presented in the microscopic images is not sufficient to check the existence or elimination of the micro-cracks?
We disagree with the reviewer's point. In our opinion, the DED of epitaxial growth of Silicon from Silicon powder onto Silicon wafers is described and presented in the results and discussion section of the paper. Concerning a "successful layer attachment", the reviewer may not be familiar with the melting behavior of silicon that shrinks by melting below the surface of the original monocrystalline Silicon wafer and as pure solid silicon floats on the melt and dissolves in it (fully or partially) there is no "adlayer" to be distinguished from the solidified melt. We did not measure the microhardness nor the mechanical properties of the silicon pillars yet. We agree that the scale of the images is not sufficient to check the existence of microcracks, but this was not our concern, because macro-cracks – detachment of the pillars occurred under most experimental conditions. Appearance of microcracks and its elimination will be looked at in the future. Nevertheless, we added the SEM pictures from which the EBSD results were obtained for showing more details of some defects in the cross sections.
3- The authors stated that “From our industrial partners from the MEMS and watch making industry an interest in 3-D printing of crystalline silicon on silicon was mentioned being of interest.”, this contradict the proposed process conditions in the current study by using a large laser spot of 600 um diameter. So, the results might be useless to the recommended applications because of the relatively small scale and dimensional accuracy needed for MEMS or watch making industry.
We agree with the statement of the reviewer that the presented dimensions are not yet interesting for our industrial partners who expressed an interest in several tens to hundred micrometer sized structures by 3-D printing of Silicon. We did not write that we achieved the needs – we just mentioned the interest and therefore our motivation to start deposition of Silicon on Silicon. We achieved macro-crack free epitaxial growth of Silicon as a first step towards the aims of our industrial partners, mentioning in the outlook the next steps to achieve this goal.
4- A design of experiment has a great role in a novel study such as the current one, what is the basic principle or reference to apply the selected range of the process parameters?
We agree with the reviewer, that design of experiment is a useful tool for finding influences of parameters on specific properties and criteria, especially in very complex multimaterials systems such as metal alloys with different microsegregation of compounds, different phases of alloys with different microstructures, in complex thermal gradient fields and cooling rates.
Silicon on the other hand is quite simple, only the cubic diamond structure, obtained at a quite precise solidification temperature is obtained, but under expansion of 10%. Cracking occurs in the cold (never melted) substrate, so the simple approach carried out was – as described in the manuscript increase the substrate temperature above the brittle to ductile transition temperature to the highest reachable temperature by the heating system at hand, and increase the temperature of the Silicon with the laser irradiation smoothly above the melting temperature by increasing the repetition rate of the laser and decreasing correspondingly the energy per pulse. Feeding in Silicon powder had different counter acting effects, that change strongly and strongly non-linearly within very small variations of parameters, therefore few experiments were carried out to find out if growth would work. To that point no DOE was needed as the number of parameters were sufficiently small to just carry out first simple series of measurements.
The other parameters were comparably logically adapted by real time high-speed camera observation of the sufficiently large melt pools for easy observation.
We agree that for the improvement and optimization of the parameters, applying a DOE approach might be very helpful but presently it was not needed.
5- The cracks formation mechanisms during the additive manufacturing process had been covered in the literature for different materials, these mechanisms might help in the current study to select the process parameters and its conditions to reduce the thermal residual stresses or avoid the as-built defects. Also, some of the data presented in the introduction section should be supported by more references and details, for example at lines #135,154. Some recent studies might be useful to enrich the introduction section such as the following for example:
- Li C, Guo YB, Zhao JB. Interfacial phenomena and characteristics between the deposited material and substrate in selective laser melting Inconel 625. Journal of Materials Processing Technology. 2017 May 1;243:269-81.
- Maamoun, A.H.; Xue, Y.F.; Elbestawi, M.A.; Veldhuis, S.C. Effect of Selective Laser Melting Process Parameters on the Quality of Al Alloy Parts: Powder Characterization, Density, Surface Roughness, and Dimensional Accuracy. Materials 2018, 11, 2343.
- Maamoun, A.H.; Xue, Y.F.; Elbestawi, M.A.; Veldhuis, S.C. The Effect of Selective Laser Melting Process Parameters on the Microstructure and Mechanical Properties of Al6061 and AlSi10Mg Alloys. Materials 2019, 12, 12.
- Lu N, Lei Z, Hu K, Yu X, Li P, Bi J, Wu S, Chen Y. Hot cracking behavior and mechanism of a third-generation Ni-based single-crystal superalloy during directed energy deposition. Additive Manufacturing. 2020 Apr 13:101228.
The cited papers all refer to solidification shrinkage cracking of alloys, that occur for these high performance alloys. We congratulate the researchers for their excellent works, but the cracks we were confronted with during the formation in our Silicon deposition process appears in the solid, never melted wafer substrate, so all hot cracking mechanisms due to shrinkage of solidifying material or low melting liquid formation in alloys are probably not relevant for our process. Remember, Silicon expands by 20% when it solidifies, it might exert quite high compressive stress (that was also measured by Raman spectroscopy and is reported in the cited reference 9, the PhD thesis of one of us. Most of our observed cracks appear during expansion of the solid silicon, before melting. Then, the solidification process is closer to classical slowly solidifying Czochralski growth of Silicon than to Ni-based superalloy solidification.
6- In the materials and methods section, some details should be included such as the laser power range of the used DED machine, and the polishing procedures.
Our DED machine is self-made and a large range of lasers can be mounted on it. For this study, we used a lamp-pumped pulsed Nd:YAG Laser with pulse durations from 0.25 ms to 500 ms pulse duration and variable energy per pulse settings. Furthermore the repetition rate of the laser can be freely selected up to 500 Hz. The maximum value of the peak power intensity of 67.5 MW/cm2 which is given in the text of the manuscript. The specifications of our Laser is 0.15 kW to 7 kW per pulse. Do you think this is an important information for the reader?
The polishing procedure with embedding, cutting, and polishing has already been mentioned in the text. If the editors consider it an important asset, we could add more details for the sample preparation process. We do not consider the need of further details of the polishing steps to reach EBSD surface quality, everybody carrying out EBSD, needs to develop the specific polishing procedure for obtaining EBSD quality. The preparation procedure has been added to the manuscript lines 119-125.
7- Fig. 2 presented the optical microscope images for the samples that were built using a feed rate of 15 g/min. Then it was mention in the text that the feed rate of 1 g/min was applied for building another, the optical microscopic images of these samples could be presented as well to clearly discuss the effect of the feed rate parameter.
Figure 2 presents the cracking of the substrate, which is function of the laser irradiation parameters and the substrate temperature mainly. Therefore, the feed rate is only marginally influencing the cracking behavior, which is one of the challenges for growing Silicon on silicon wafers. Therefore, it is shown here. The topic of this communication lies on epitaxial growth of Silicon that works only under low feed rates, as mentioned in the text. One high feed rate result, without crack, is now shown in Figure 3 c, where the epitaxial growth is not maintained under the otherwise identical growth conditions. In a DOE, to be carried out, the maximum feed rate for epitaxial growth, together with the corresponding other parameters will be determined, but this has not been done yet. We could also present the full PhD thesis in this letter, but this would result in about 70 pages, with 20 reviewers each arguing that other details would need to be studied.
8- The conclusion section still needs to be re-organized in a simple way to easily track the main results and contributions presented in the current study.
The conclusion has been re-organized.
Please revise the manuscript carefully by considering the above questions and comments.
We thank the reviewer for taking the time again to read and accept our aim of presenting in a succinct communication our interesting first results of growing Silicon on shrinking to melt Silicon wafers.
Reviewer 3 Report
This manuscript mentioned that an attempt of epitaxial growth of Si wafers by selective laser melting of Si powders. The revised manuscript is sufficient interest for publication.
Author Response
We thank the reviewer for having accepted our paper.